# Troponin Cut-Offs for Acute Myocardial Infarction in Patients with Impaired Renal Function—A Systematic Review and Meta-Analysis

**DOI:** 10.3390/diagnostics12020276

**Published:** 2022-01-21

**Authors:** Jan Kampmann, James Heaf, Christian Backer Mogensen, Andreas Kristian Pedersen, Jeff Granhøj, Hans Mickley, Frans Brandt

**Affiliations:** 1Internal Medicine Research Unit, University Hospital of Southern Denmark, Sydvang 1, 6400 Sonderborg, Denmark; Jeff.Granhoj2@rsyd.dk (J.G.); FBK@rsyd.dk (F.B.); 2Department of Regional Health Research, University of Southern Denmark, Campusvej 55, 5230 Odense, Denmark; Christian.Backer.Mogensen@rsyd.dk (C.B.M.); Andreas.Kristian.Pedersen@rsyd.dk (A.K.P.); 3Department of Medicine, Zealand University Hospital, Sygehusvej 10, 4000 Roskilde, Denmark; james@heaf.net; 4Department of Emergency Medicine, Hospital of Southern Jutland, Kresten Philipsens Vej 15, 6200 Aabenraa, Denmark; 5Department of Cardiology, Odense University Hospital, J. B. Winsløws Vej 4, 5000 Odense, Denmark; Hans.Mickley@rsyd.dk

**Keywords:** acute myocardial infarction, chronic kidney disease, troponin

## Abstract

Identifying acute myocardial infarction in patients with renal disease is notoriously difficult, due to atypical presentation and chronically elevated troponin. The aim of this study was to identify a specific troponin T/troponin I cut-off value for diagnosis of acute myocardial infarction in patients with renal impairment via meta-analysis. Two investigators screened 2590 publications from MEDLINE, Embase, PubMed, Web of Science, and the Cochrane library. Only studies that investigated alternative cut-offs according to renal impairment were included. Fifteen articles fulfilled the inclusion criteria. Six studies were combined for meta-analysis. The manufacturer’s upper reference level for troponin T is 14 ng/L. Based on the meta-analyses, cut-off values for troponin in patients with renal impairment with myocardial infarction was 42 ng/L for troponin I and 48 ng/L for troponin T. For patients on dialysis the troponin T cut-off is even higher at 239 ng/L. A troponin I cut-off value for dialysis patients could not be established due to lack of data. The 15 studies analyzed showed considerable diversity in study design, study population, and the definition of myocardial infarction. Further studies are needed to define a reliable troponin cut-off value for patients with kidney disease, especially in dialysis patients, and to allow necessary subanalysis.

## 1. Introduction

According to the fourth definition of myocardial infarction, the three diagnostic pillars for acute myocardial infarction (AMI) include typical symptoms, typical ECG changes, and a fall or rise in cardiac enzymes—preferably cardiac troponin (cTn) with one of the cTns being above the 99th percentile of the upper reference limit (URL) [1]. Renal impairment is known to be strongly associated with increased risk of cardiovascular disease [2]. The fourth universal definition of myocardial infarction (UDMI) suggests similar criteria for diagnosing AMI among patients with chronic kidney disease (CKD) as those for the general population [1]. However, both symptoms of AMI and changes in ECGs are different in CKD patients in comparison with the background population. The most common symptom associated with AMI in CKD patients is dyspnea and not chest pain [3]. Also, ECG abnormalities, such as left bundle branch block (LBBB), right bundle branch block (RBBB), and left ventricular hypertrophy (LVH) are more frequent in patients with CKD [4]. Non-ST-elevation myocardial infarction (NSTEMI), which is highly dependent on cTn values, is 2–4 times more common in CKD patients [3,5]. ST-elevation myocardial infarction (STEMI) is based on ECG changes that may be obscured by LBBB, RBB, LVH, and hyperkalemia-related ECG changes. Combined with atypical symptoms, physicians depend to a higher degree on cTn measurements.

CTn is often elevated in patients with CKD per se, and the concentration increases with decreasing renal function [6]. Furthermore, the positive predictive value (PPV) of highly sensitive cardiac troponin I (Hs-cTnI) for AMI in CKD patients decreases with decreasing glomerular filtration rate to 15–32% in dialysis patients [7]. Pfortmueller et al. concluded that the diagnostic value of highly sensitive cardiac troponin T (hs-cTnT) for diagnosing AMI in CKD patients was comparable to a toss of a coin [8]. Consequently, only about 20% of CKD patients with AMI are diagnosed correctly at hospital admission, as opposed to about 40% of non-CKD patients [3].

An initial correct diagnosis at hospital admission is essential for both STEMI and NSTEMI patients. Timely management and coronary arteriography (CAG) are pivotal in patients with STEMI [9]. Delay in coronary angiography and intense dual-antiplatelet therapy has been shown to increase mortality in NSTEMI patients [10].

A reliable cut-off value for cTn in CKD patients with AMI, therefore, may be crucial in order to optimize early detection of AMI and thereby timely initiation of treatment.

The available literature on specific cTn cut-off values for AMI (STEMI and NSTEMI) in adult patients with renal impairment was reviewed. Meta-analysis was employed to establish a CKD-specific cTn cut-off for AMI.

## 2. Materials and Methods

### 2.1. Protocol and Registration

The protocol was registered on 28 April 2020 (ID CRD42020162299).

### 2.2. Eligibility Criteria

We included studies evaluating cTn in adult patients with suspected renal impairment for AMI. Only articles in English were considered eligible. No limitations as to publication date were set.

Inclusion criteria: age above 18 years; only full-text articles in peer-reviewed journals; only studies that investigated alternative cut-off values according to renal impairment.

Exclusion criteria: posters or abstracts; letters to the editors; and other grey literature. 

Information sources:

Embase: last search 18 February 2020

MEDLINE: last search 18 February 2020

PubMed: last search 17 February 2020

Cochrane: last search 3 February 2020

Web of Science: last search 3 February 2020

The search procedure included the following search engines and websites: Embase, MEDLINE, PubMed, Cochrane Library, and Web of Science. The search strategy was assisted by a specialized librarian and can be seen under Appendix A. 

### 2.3. Study Selection

Preliminary selection of articles was performed by titles and abstract, followed by a detailed full text selection based on inclusion and exclusion criteria. Assessment was executed by Jan Dominik Kampmann (JDK) and Jeff Granhøj (JG). In the case of disagreement, a referee Frans Brandt (FB) was consulted. The online systemic review manager program Covidence™ was used during the extraction period.

### 2.4. Data Extraction

Using a standardized data extraction form investigator, JDK extracted relevant details and results. The year of publication, study population, study design, and definition of renal impairment are listed in Table 1, and cut-off-value-related data in Table 2. The extracted data were verified by JG.

### 2.5. Methodological Quality Assessment

Quality and bias were assessed by two independent investigators, JDK and JG, using the Quadas2 score. The Quadas2 score is a validated tool for quality assessment of diagnostic accuracy [11]. The tool content was tailored by exchanging the domain “Index test” with reporting bias defined by how much detail the cTn measurement was described in (detection limit, manufacture, 99th percentile) and how well renal impairment assessment was described (estimated glomerular filtration rate (eGFR) formula, assay used for creatinine measurement). In cases of disagreements regarding the QUADAS-2 results, consensus was reached by discussion between JDK and JG. If no agreement could be reached, FB was consulted as third reviewer.

### 2.6. Statistical Analysis

A bivariate mixed-effect model on the sensitivity and specificity transformed by way of the inverse probit function, similar to the model implemented in the R-package diagmeta was employed in order to calculate the optimal cut-off value in accordance with the area under the curve (AUC) for cardiac troponin T(cTnT) and cardiac troponin I(cTnI) in patients with renal impairment [12]. We chose this model because several studies had multiple and varying numbers of cut-off values with corresponding sensitivity and specificity. The AUC showed that the optimal link function was probit for all studies, and the optimal cut-off value was estimated by way of the summary receiver operating characteristic (SROC) curve. Therefore, the optimal cut-off point maximizes the area under the curve of the SROC curve, which is an estimate of the true underlying ROC curve of the studies in the analysis. We assumed that the covariance structure had no random intercept and a common random slope. The choice for this structure was based upon that different cut-off values that were calculated on different patients and a more complex model was unable to converge. A SROC curve for cTnT and cTnI stratified for dialysis for all studies was utilized to assess bias, because challenges in interpretation of funnel plots arise when each study has multiple cut-off values and corresponding sensitivity and specificity. The optimal cut-off points, their corresponding sensitivity and specificity, and their 95% confidence intervals were calculated as described in [12]. The analysis was done in Rstudio™ with the R-package diagmeta.

## 3. Results

As our paper is a systematic review on troponin cut-off values in CKD patients and a meta-analysis, results and discussion are divided into a narrative synthesis and a meta-analysis in order to improve readability.

### 3.1. Narrative Synthesis

A total of 2590 publications were screened. A flowchart according to “Referred Reporting Items for Systematic Reviews and Meta-Analyses: the PRISMA Statement 2009” is presented in Figure 1 [13]. During the title and abstract screening, 2544 studies did not fulfill the inclusion criteria. Publications in other languages than English and publications not dealing with cTn cut-off values in patients with AMI and renal impairment were excluded. The remaining 46 publications were again assessed for eligibility according to inclusion and exclusion criteria. A total of 15 articles published from 2007 [14]–2019 [15], were preliminarily included. A total of six studies, which were considered to be adequate in terms of AMI and renal impairment definition, were selected for the meta-analysis.

In the following all 15 studies are presented in combination. Studies featured in the meta-analysis and the meta-analysis itself are discussed in detail at the end of the result section. Study characteristics of the included studies can be seen in Table 1.

### 3.2. Study Design

The 15 studies included for review featured four retrospective studies [15,16,17,18], eight prospective studies [14,19,20,21,22,23,24,25], and three post hoc analyses [26,27,28].

One of the four retrospective studies was a multicenter study [18]. Three of the eight prospective studies were multicenter studies [23,24,25]. Two of the post hoc analyses were based on a multicenter study [27,28]. The other studies were single-center studies [14,15,16,17,19,20,21,22,26].

### 3.3. Renal Impairment

In the majority of studies, renal impairment was defined as estimated glomerular filtration rate (eGFR) < 60 mL/min/1.73 m^2^ [17,19,20,21,22,23,24,25,27,28]. Soeiro et al. included patients with a creatinine level >1.5 mg/dL [16], and Canney et al. included patients with an eGFR < 45 mL/min/m^2^ [18]. This cohort study was designed to predict death, hence the lower eGFR. Two publications included only ESRD patients on dialysis [15,26].

In order to estimate GFR, the chronic kidney disease epidemiology collaboration equation (CKD-EPI formula) [17,18,22,25,28], the Japanese equation formula [20], The modification of diet in renal disease equation (MDRD) [19,21,23,24,25,27], and the Cockcroft–Gault formula were used [14,21]. Chotivanawan et al. used both the Cockcroft–Gault and the MDRD formulas [21].

Five studies have subdivided renal impairment into CKD stages 3–5 according to KDIGO guidelines [17,18,20,21,22]; however, only Iwasaki et al. included patients with CKD stage 1–2 [20].

### 3.4. Identification of Patients Suspected Having an AMI

Patients most often were suspected of having AMI based on symptoms [14,15,16,17,19,20,22,23,24,25,26,27]. Kraus et al. defined the measurement of at least two cTn during the study period as surrogate markers for suspicion for AMI [28].

Chotivanawan et al. [21] and Canney et al. [18], included asymptomatic patients in order to establish a mean cTn value in asymptomatic renal impairment patients or to establish the long-term risk of cardiovascular events, respectively.

### 3.5. Defining the Endpoint AMI

There was a heterogeneous definition of AMI in the included articles. Four articles used NSTEMI as an endpoint [16,23,24,28], Miller-Hodges et al. included cardiovascular death within 30 days additionally to NSTEMI diagnosis [23].

Seven articles used AMI defined as NSTEMI and STEMI as endpoints [14,15,17,20,22,25,26,27]. Only Chenevier-Gobeaux et al. looked additionally for different cut-off values for cTn for NSTEMI and AMI (STEMI and NSTEMI combined), respectively [27].

Three articles demanded additionally a significant coronary lesion ≥70% [16,22] and >50% respectively [20], visible on on coronary angiography, to diagnose AMI.

Two articles chose different endpoints. Canney et al. established a cut-off for CV risk in asymptomatic CKD patients [18]. Chotivanawan et al. suggested a cut-off based on the mean hs-cTnT in asymptomatic patients with CKD stage 3–5 [21].

### 3.6. Applied Consensus Documents Regarding Definition of the Diagnose AMI

Seven articles specified the consensus documents used for the diagnosis of AMI.

Sukonthasarn et al. and Ryu et al. refer to the 2000 “Myocardial Infarction Redefined—A Consensus Document of The Joint European Society of Cardiology (ESC) and the American College of Cardiology Committee for the Redefinition of Myocardial Infarction (ACCF)” [14,26]. Lim et al. refer to the fourth definition of MI from 2018 by ESC, ACCF, the American Heart Association (AHA), and the World Heart Federation Task Force (WHF) [15]; Yang et al. and Chenevier-Gobeaux et al. to the third universal definition of myocardial infarction from 2012 by ESC, ACCF, AHA, and WHF [17,27]; and Flores-Solis et al. [19] and Twerenbold et al. [24] to the ACS consensus documents by the ESC, ACCF, AHA, and WHF from 2007. Iwasaki et al. used the ESC, American College of Cardiology, American Heart Association and World Heart Federation Task Force consensus document [20].

### 3.7. Confirmation of the Diagnosis Myocardial Infarction

Out of the 13 publication that had ACS or AMI as endpoint, six authors specified by whom the diagnosis of AMI/ACS was suggested [15,17,20,25,27,28].

Out of the 13 publications, 6 used a varying number of cardiologists to confirm the diagnosis AMI/ACS [15,17,20,25,27,28]. In two articles the cardiologists were blinded for the cTn result [20,28]. Kraus et al. used two cohorts in their study. In one of the cohorts, the AMI diagnosis was based on the diagnosis at hospital discharge, in the other cohort the AMI diagnosis was made by two independent cardiologists [28].

### 3.8. Patient Characteristics

The proportion of men with renal impairment in the study population ranged from 34.8% [14] to −68% [19]. The age of patients with renal impairment in the studies was between 60.9 [26] and 79.0 years [24]. One article did not report the age and sex distribution of participants with renal impairment [27]. The size of study population ranged from 46 [14] to 2284 patients [26].

### 3.9. Quality Assessment

In order to determine the risk of bias and applicability of the individual studies, the QUADAS-2 score was applied. Results are shown in Figure 2 and Figure 3.

The risk for bias was low in the studies in general. Only two studies [14,22] scored “unclear” in terms of reporting bias and none of the studies had a high risk for reporting bias or patient flow according to QUADAS-2 score. Patient selection was more problematic with three studies [14,16,17], having a high risk for bias and four studies [15,20,24,28] with an unclear risk of bias. Three studies [16,18,19] were considered high risk in terms of reference standard bias, ten [14,15,17,20,22,23,24,25,27,28] low risk, and two studies [21,26] were regarded as unclear.

The results regarding concerns of applicability were less convincing. Reporting bias were of less concern with 12 [14,15,16,17,19,20,23,24,25,26,27,28] low-risk and three [18,21,22] high-risk articles. However, the reference standard yielded five [16,18,19,21,22] high-risk studies concerning applicability and seven [14,15,17,20,25,27,28] low-risk studies. Applicability in terms of patient selection showed five high-risk [14,15,16,18,21] and six low-risk studies [17,19,20,23,25,27].

Chenevier-Gobeaux et al. [27] and Twerenbold et al. [25], were the best studies according to our evaluation in terms of risk of bias and concerns regarding applicability, with low-risk scores in all categories. The study with most concerns regarding applicability was by Canney et al. [18], with high risk scoring throughout. Risk of bias was highest for the study by Sukanthasarn et al. [14], with only one low-risk score in reference standard, high-risk inpatient selection, and unclear results in reporting bias and flow and timing.

### 3.10. Statistical Analysis across Studies

Most studies calculated cut-off values for cTn using ROC [14,15,16,17,19,20,22,24,25,26,27,28]. Canney et al. used the repeated log-rank test [18], while Miller-Hodges et al. calculated the negative predictive value and sensitivity for hs-cTnI < 5 ng/L [23] and Chotivanawan et al. calculated the mean cTn of their cohort [21].

### 3.11. A New Cut-Off

In the relevant studies defining AMI as endpoint [14,15,16,17,19,20,25,26,27,28], the cTn cut-off values varied from 25.9 ng/L [25] to 350 ng/L [26] according to CKD stage and cTn assay. In terms of relative change, the cTn cut off values differed from 0.85 times the upper reference limit (URL) [25] to 12 times the URL [16]. Two cTnI assays have shown a lower optimal cut-off value than the suggested 99th percentile/URL by the manufactures [25].

Two articles looked exclusively at end stage renal patients (ESRD) patients on dialysis resulting in a higher cut-off value [15,26]. Yang et al. showed a lower cut-off value for cTn in the CKD stage 5 group not on dialysis compared to CKD 4 patients. Soeiro et al. suggest a higher cut-off value in patients without renal failure compared to patients with renal impairment (6.05 ng/L compared to 5.20 ng/L) [16]. Iwasaki et al. used a four-pattern semiquantitative measurement of cTnT, where only changes between 0.1–2.0 ng/mL were displayed numerically [20].

The sensitivity of cut-off values from the relevant studies ranged from 55.6 (corresponding specificity 92.2) [20] to 95 (corresponding specificity 97) [26]. Specificity ranged from 60.76 (corresponding sensitivity 93.3) [15] to 97 (corresponding sensitivity 95) [26]. Ryu et al. provided the highest combined specificity and sensitivity, as shown in Table 2 [26]. Troponin cut-off details can be seen in Table 2.

### 3.12. Meta-Analysis

Six papers [15,17,25,26,27,28] were eligible for meta-analysis defined by relevant study population, AMI definition, and comparable statistical method. The respective studies provided the following amount of cut-off values in total: Gobeaux et al.—*n* = 3, Kraus et al.—*n* = 3, Lim et al. —*n* = 3, Ryu et al.—*n* = 11, Twerenbold et al.—*n* = 35, and Yang et al. —*n* = 2.

A total of 15 studies were included for the qualitative synthesis and of these, six studies were included in the meta-analysis. Three articles were excluded as the authors only accepted the diagnosis myocardial infarction when a significant coronary artery occlusion was present on angiography [16,20,22]. One study was not eligible for meta-analysis due to the inclusion of unstable angina pectoris as endpoint [19]. Two articles only included asymptomatic patients and were excluded [18,21]. Two other studies were not considered for meta-analysis since both tested a cut-off value from a previous study [23] or an alternative algorithm [24]. The Sukanthasarn et al. [14] article was excluded due to high risk scores in applicability and risk of bias in the QUADAS-2 assessment.

The curve was computed by gathering the provided cut-off values from the different studies together with the respective sensitivity and specificity.

The meta-analysis established that the optimized cut-off for cTnI and cTnT from all the six studies included is at 73 (25.75; 119.45). The sensitivity and specificity of the cut-off values were 0.78 and 0.84, respectively, with an AUC of 0.89 (see Figure 4).

For patients with eGFR < 60 mL/min/1.73 m^2^ not on dialysis, the optimized cut-off for cTnT according to meta-analysis was at 48 ng/L (23.95; 71.83). Four studies were included, providing 11 cut-offs. Kraus et al., Twerenbold et al., Gobeaux et al., and Yang et al. provided 2, 5, 3, and 1 cut-offs respectively. All studies used a hs-cTnT assay. The sensitivity and specificity for the cut-off were 0.76 and 0.78 respectively with an AUC of 0.85, see Figure 5.

The cut-off value for cTnT for patients on dialysis was established at 240 ng/L (69.27; 410.23). The analysis consisted of 2 studies where Ryu provided 11 cut-off values and Yang 1 cut-off value. Ryu et al. used a cTnT, whereas Yang used an hs-cTnT assay. The sensitivity and specificity for the cut-off values were 0.88 and 0.92 respectively with an AUC of 0.96. See Figure 6.

The cTnI cut-off for patients with eGFR < 60 mL/min/1.73 m^2^ not on dialysis was established at 42 ng/L (33.83; 51.08). Kraus et al. provided one cut-off and Twerenbold 30. The cut-off for cTnI for patients on dialysis could not be analyzed as only one study using troponin I included dialysis patients [15]. Kraus et al. used a hs-TnI assay. Twerenbold used three cardiac cTnI and three hs-cTnI assays. The sensitivity and specificity for the cut-off were 0.77 and 0.86 respectively with an AUC of 0.89. See Figure 7.

Table 3 shows all the different estimated cut-offs with the respective sensitivity, specificity, AUC, and CI values.

## 4. Discussion

### 4.1. Narrative Synthesis

The interpretation of cTn in CKD patient with AMI is challenging. The prevalence of chronically elevated cTn is high and especially the first cTn measurement is difficult to interpret. While waiting for a second cTn to establish dynamic changes, relevant interventions may be delayed. 

The studies included in this review were heterogeneous in terms of definition of renal impairment and definition of AMI.

The definition of renal impairment varied due to the different eGFR formulas used. The definition of chronic kidney disease opposed to a random eGFR measurement was vague in many of the included studies. It has not been established whether acute renal failure does have a different impact on cTn compared to chronic kidney disease and might have had an impact on the results.

No study presented information on dialysis details or as to the timing when the cTn sample was obtained. Studies have shown different results on the influence of dialysis on cTn [29,30]. Without those details, the cTn measurements may be unreliable. The cut-off span from 75 ng/L for hemodialysis (HD) patients to 144 ng/L for the seven PD patients included in the study was wide. PD patients tended to have higher baseline cTn [31]; however, a twice-as-high cut-off seems extreme.

Only three studies [17,18,20] provided different cut-off values for cTn according to different stages of CKD. The cut-off values showed considerable diversity. Only Chenevier-Gobaux et al. described that the area under the curve in their study did not vary according to eGFR categories; however, only 75 patients with CKD were involved in the study [27].

One study showed the following cut-off values: CKD stage 1–2: 47 ng/L; stage 3: 88 ng/L; stage 4–5 180 ng/L, and for patients on dialysis 270 ng/L [20].

The suggested cut-off values were considerably higher in studies with a predominantly Asian study population [17,20]. In contrast to the study by Yang et al. [17] who suggested a cut-off level for hs-TnT of 129 ng/L, Twerenbold [25] et al. suggested a cut-off level of 29 ng/L when using the same assay. Iwasaki et al. proposed a cut-off for CKD 4–5 patient not on dialysis of 180 ng/L. The differences might reflect ethnic differences in cTn.

Yang et al. argued that 88% of Twerenbold’s study population were in CKD stage 3, whereas there were only approximately 30% CKD 3 patients in Yang et al.’s study population. A subdivision into the different CKD stages would have yielded important information and is therefore desirable in future studies.

The articles included in the review used different definitions for diagnosing AMI. The definition of AMI has changed significantly from 2000 to 2007. One article combined two cohorts, which used different definition of AMI as endpoint [28]. This might have influenced the final diagnosis.

NSTEMI is more common in CKD patients than STEMI [3,5]. The ECG changes and symptoms can cause problems in CKD patients due to atypical presentation, making NSTEMI hard to diagnose, and leaving the physician dependent on cTn results [4,32,33]. The diagnostic dilemma has been shown in Twerenbold et al.’s study in which the two independent cardiologists disagreed more frequently when it came to diagnosing NSTEMI in patients with renal impairment in comparison to patients with normal eGFR (13.1% vs. 9.1%, *p* = 0.006) [24]. It is therefore debatable if NSTEMI as endpoint is reliable.

NSTEMI is a clinical diagnosis and coronary angiography is not mandatory [34]. Previous studies have shown that 40–50% of patient have coronary stenosis before starting dialysis [35] causing potential bias when using coronary occlusion as an endpoint as was done in some studies.

Chenevier-Gobeaux et al. underlined the importance of age as influential factor on cTn alteration [27]. The authors propose that an adapted threshold for cTn is required for patients with low eGFR and ≥70 years of age [27]. Kraus et al. presented an algorithm based on admission cTn and dynamic changes [28]. Flores-Solís et al. suggested including CK-MB, which improved the diagnostic accuracy in their study [19]. Twerenbold et al. tested the recently proposed 0/1-h European Society of Cardiology (ESC) algorithm for patients with low eGFR [24]. The 0/1-h algorithm is designed for patients with NSTEMI and is based on cTn concentrations at presentation and their absolute change after 1 h. Using slightly higher cut-offs for ruling out NSTEMI yielded only a small improvement in their study and the authors did not recommend altering the cut-off for the 0/1-h algorithm for patients with low eGFR [24].

### 4.2. Meta-Analysis

According to our meta-analysis, the cTnI cut-off lies at 42 ng/L and at 48 ng/L for cTnT. For patients on dialysis the cTnT cut-off is as high as 240 ng/L.

The calculated cut-off values for cTn in patients with renal impairment and myocardial infarction were in general higher than the URL suggested by the assay manufacturers.

The manufacturer’s upper reference level for cTnT is 14 ng/L. The suggested cut-off for cTnT is 3.4 times higher than the URL for non-dialysis patients and 17 times higher for dialysis patients.

In cTnI the URL ranges from 9 (TNI Siemens) to 42 ng/mL (TNI Beckman Coulter), corresponding to a cut-off 4.7 times higher than, or roughly identical to, the manufacturer’s URL.

The sensitivity and specificity of the generated cut-offs were generally high, with a sensitivity of 0.76 for TnT in patients without dialysis and 0.88 in dialysis patients respectively. The corresponding specificity was 0.78 and 0.92 respectively. For cTnI cut-off in non-dialysis patients, the sensitivity was 0.77 and specificity 0.86. This gives us some confidence that our cut-offs are clinically useful. However, it should be noted that only the cTnT cut-off value for non-dialysis patients was based exclusively on highly sensitive assays. The other meta-analysis consisted of data were from both highly sensitive and less-sensitive assays.

### 4.3. Limitations

There are many conditions that cause elevated cTn levels [21,36]. Both the sensitivity and specificity of cTn can be affected by various factors, such as the respective assay, manufacturer, sex, and the age of the patient [7,25,27,37,38]. A within-day and between-day reference change from 46% to −32% and from 81% to −45% respectively in cTnI have been described [39]. Due to this, an optimized cut-off should therefore always be interpreted with caution.

Our meta-analysis depended on multiple suggested cut-off values from the different studies. Most of these cut-offs derive from Twerenbold et al. [25]. However, this study scored highly in the QUADAS-2 quality score, emphasizing its quality, and are in line with cut-offs from the other high-scoring study by Chenevier-Gobeaux [27]. The featured TnI assays in Twerebold’s work had a wide range of 99th percentile from 9 ng/L to 42 ng/L, yet the optimized cut off-range was much narrower, ranging from 26 ng/L to 46 ng/L [25]. In order to produce meaningful troponin cut-off values, our study includes the current hs-Tn as well as cTn assays. However, although the 99th percentile values may be lower for highly sensitive assays, the cut-off values were similar in high cTn and hs-Tn assays [25]. Therefore, we argue that using both assays does not alter the quality of the cut-off value.

We acknowledge that an assay specific cut-off value and a subanalysis for NSTEMI is crucial. This, however, was not possible with the included studies.

The cTnI cut-off for patients not on dialysis and the cTnT cut-off for patients on dialysis is based on only two studies. The analysis was only possible due to the several cut-offs provided by the respective studies and the respective sensitivity and specificity analysis. Several subanalyses could not be performed due to lack of data, including differences of cut-off values according to cTn versus hs-cTn assays, differences in the respective CKD stages, and differences in cut-off values in HD patients and PD patients.

## 5. Conclusions

Our review consisted of highly diverse studies. We suggest cut-offs for cTnI of 42 ng/L, and for cTnT of 48 ng/L for non-dialysis patients with eGFR < 60 mL/min/1.73 m^2^. For patients on dialysis, we suggest a cTnI cut-off of 240 ng/L. However, these cut-offs were only based on two studies, providing a substantial risk of bias. Yet, the high sensitivity and specificity of the cut-offs emphasize the validity of the optimized cut-offs. We suggest further studies with a high number of patients and homogenous definition of renal impairment and AMI to confirm or modify the findings. Further subdivision according to eGFR would be desirable in order to optimize and personalize cTn cut-offs, especially for dialysis patients. A stratification for NSTEMI vs. STEMI would provide important information. The limitations and challenges of our study can be seen as an inspiration to improving future study designs on troponin cut-off values. Trials comparing multiple highly sensitive assays are desirable, since all assays have their own individual cut-of value. In dialysis patients, cTn cut-offs should be subdivided into HD and PD. Optimized cTn cut-off values can never stand alone. Clinical assessment and thorough anamneses will always be pivotal for diagnosing AMI. Therefore, the cut-off values presented in our study should be regarded as a suggestion rather than a final conclusion, and we underline the importance of further studies on the subject.

## Figures and Tables

**Figure 1 diagnostics-12-00276-f001:**
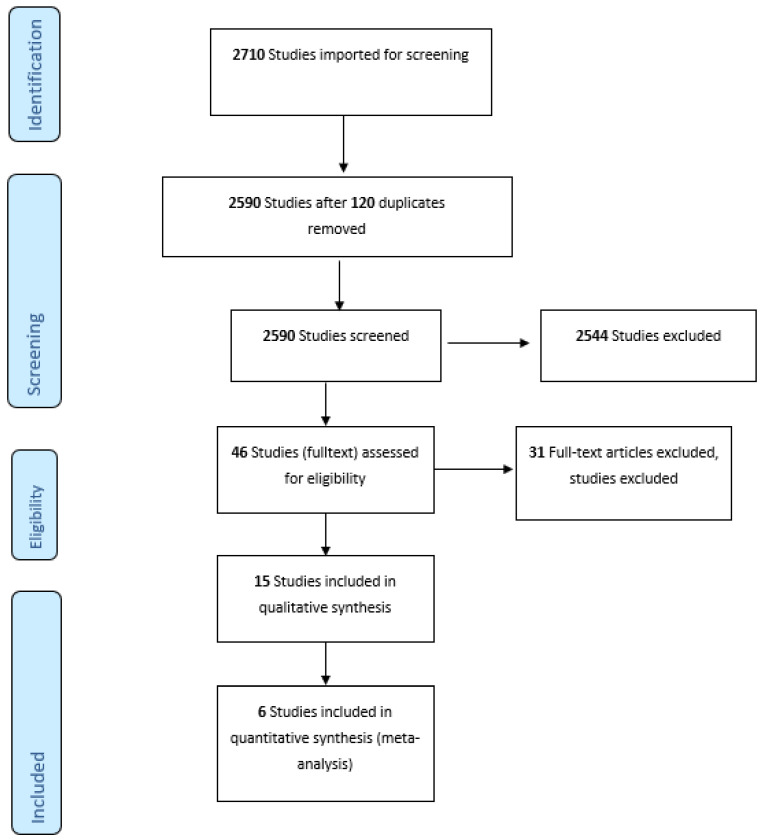
Prisma Flow chart of the screened articles for the systematic review of cut-off values for troponin in patients with chronic kidney disease.

**Figure 2 diagnostics-12-00276-f002:**
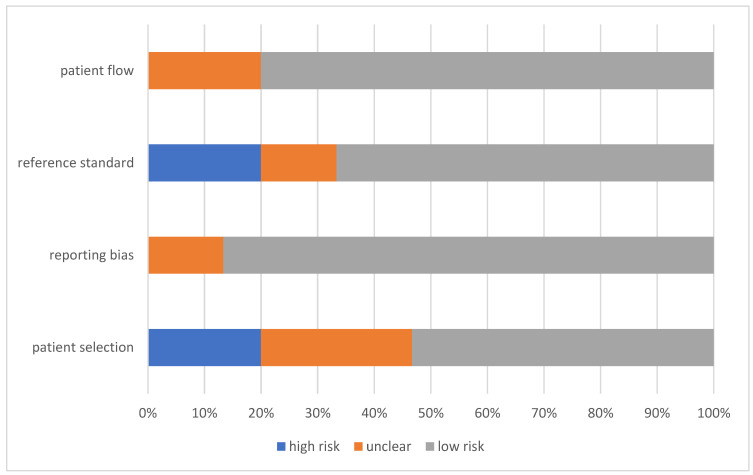
QUADAS-2 score for the risk of bias across all studies for the included studies in the systematic review of cut-off values for troponin in patients with chronic kidney disease.

**Figure 3 diagnostics-12-00276-f003:**
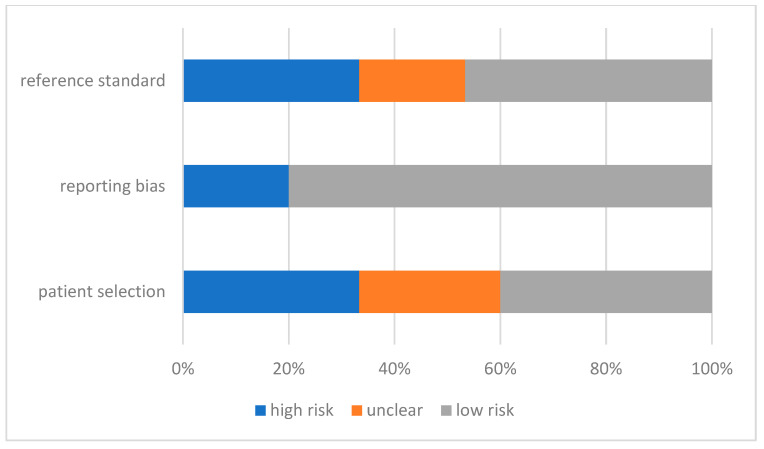
QUADAS-2 score for applicability of bias across all studies for the included studies in the systematic review of cut-off values for troponin in patients with chronic kidney disease.

**Figure 4 diagnostics-12-00276-f004:**
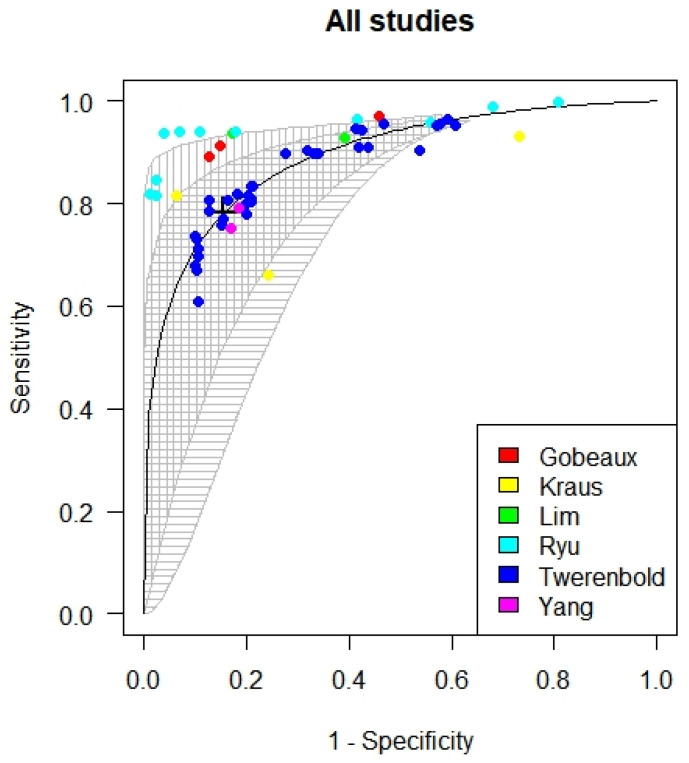
SROC curve for the optimized troponin cut-off in patients with chronic kidney disease.

**Figure 5 diagnostics-12-00276-f005:**
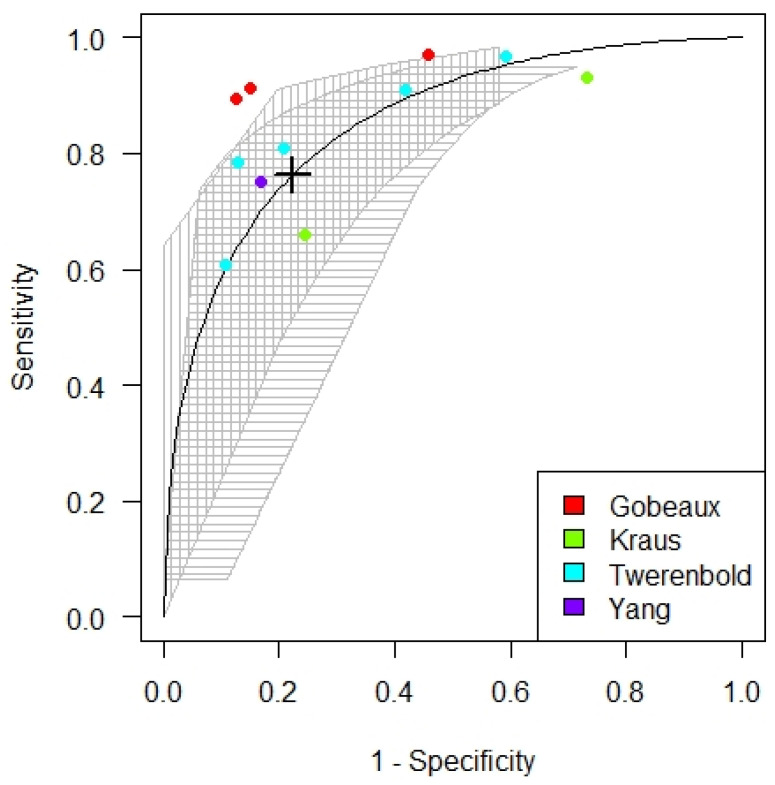
SROC curve for the optimized troponin T cut-off for patients not on dialysis.

**Figure 6 diagnostics-12-00276-f006:**
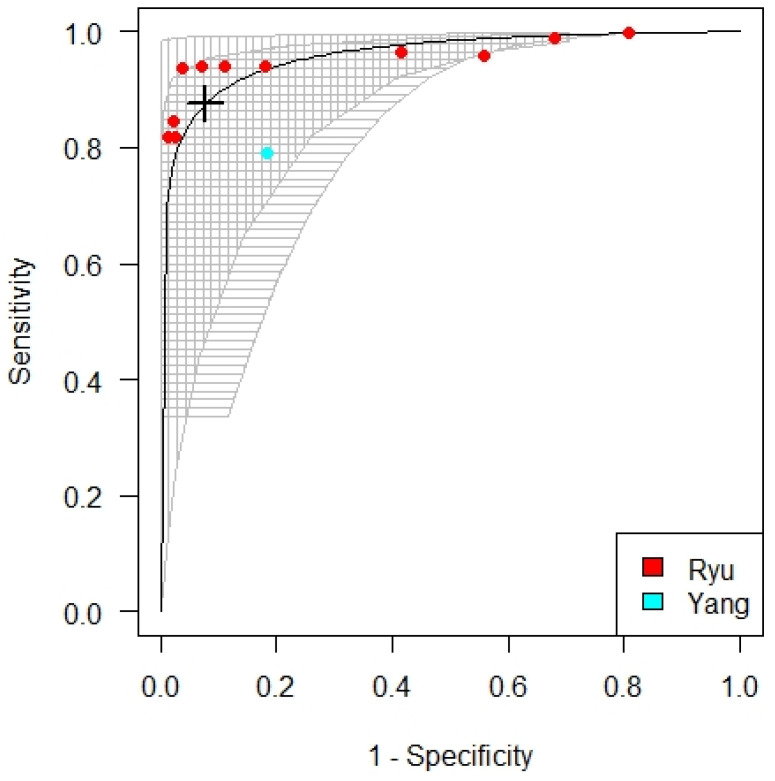
SROC curve for the optimized troponin T cut-off for patients on dialysis.

**Figure 7 diagnostics-12-00276-f007:**
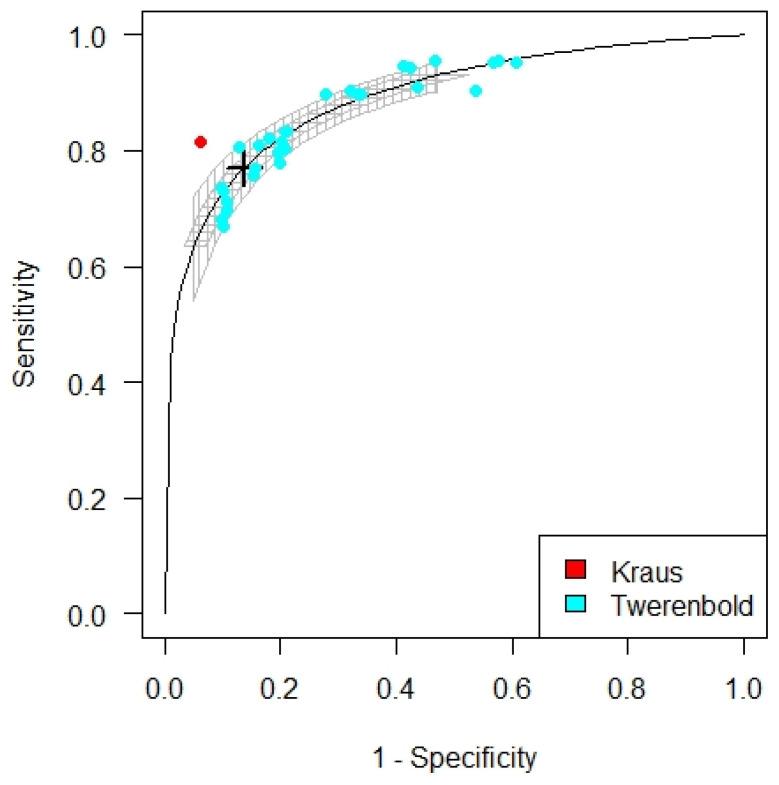
SROC curve for the optimized troponin I cut-off for patients without dialysis.

**Table 1 diagnostics-12-00276-t001:** Study characteristics of the included studies for the systematic review of cut-off values for troponin in patients with chronic kidney disease.

Author	Year	Country	Number of Patients with Renal Impairment	Renal Impairment Definition	Male Patients with Renal Impairment %	Age	Study Design	Inclusion criteria	Exclusion Criteria
Soeiro	2017	Brazil	184	Creatinine > 1.5 mg/dL	52	Median 63 years (for patients with and without renal impairment)	Retrospective, single-center, observational	Patients who presented with chest pain at the ED and underwent coronary angiography	Presence of ST elevation
Sukonthasarn	2007	Thailand	46	Crcl < 60 mL/min Cockcroft–Gault for at least 3 months	38	Mean 72 (AMI group) 70 (control group)	Cross-sectional case control	CKD and ACS symptoms	Patients with pulmonary embolism, muscle disease, acute stroke, renal dysfunction of duration less than 3 months, recent ACS other than this admission, history of recent exercise, muscle trauma, or those being treated by electrical cardioversion, and patients in the AMI group who did not fulfill the AMI criteria
Chenevier_Gobeaux	2013	France	75	MDRD eGFR > 60 mL/min/1.73 m^2^	65 in the entire study population	Mean: 57 in the entire study population	Post hoc analysis of two previous (prospective) studies	Patients presenting to ED or cardiology unit with a suspected diagnosis of AMI (chest pain onset < 6 h).	Patients requiring renal replacement therapy
Kraus	2018	Germany	1861 (1581 from data warehouse and 280 from stenocardia cohort)	EPI formula eGFR < 60 mL/min/1.73 m^2^	Stenocardia: male 58; data warehouse 56	Mean stenocardia: 72; data warehouse: 77	Post hoc analysis of 2 previous studies (1 prospective, 1 retrospective)	Stenocardia: patients with acute chest pain Data warehouse: patients with at least two measurements of hs-cTnT	Stenocardia: major surgery or trauma, pregnancy, IV abuse, anemia, dialysis, STEMI data warehouse: obviously erroneous data
Ryu	2011	Korea	284	ESRD patients on PD or HD	52	Mean 61	Post hoc analyses	Dialysis patients with ACS symptoms	Myositis, muscle trauma, rhabdomyolysis and seizure
Lim	2019	Korea	1144	82 with ESRD + MI (75 HD and 7 PD patients)	58	Median 61	Retrospective, single-center, observational ED	HD or PD, TNI measured	Insufficient medical record (with neither echocardiography nor coronary angiography) and starting dialysis at the time of admission
Yang	2017	China	489	EPI-formula CKD eGFR > 60 mL/min/1.73 m^2^ diagnose CKD confirmed by two nephrologists according to KDIGO guidelines	57	Mean 71	Retrospective, single-center study	CKD patients with chest pain and tested hs-TnT with an onset or peak within the last 12 h	Previous myocardial infarction, hypertensive crisis, tachy- or bradyarrhythmias, pulmonary embolism, severe pulmonary hypertension, myocarditis, acute neurological disease, aortic dissection, aortic valve disease or hypertrophic cardiomyopathy, cardiac contusion, ablation, pacing, cardioversion, endomyocardial biopsy, hypothyroidism, apical ballooning syndrome, drug toxicity, burns, rhabdomyolisis, and multiple organ failure
Flores-Solís	2012	Spain	484	MDRD > 60 mL/min/1.73 m^2^	68	Mean 77	Prospective single-center study	CKD3-5, sought hospital due to suspected ACS, whose initial clinical evaluation included measurement of cTnI, CK-MB and creatinine, clinical history ECG, physical examination	Patients transferred to another hospital, psychiatric patients, those who refused informed consent
Canney	2019	Canada	1956	EPI formula eGFR > 45 mL/min/1.73 m^2^	63.1	mean 68.1	Retrospective cohort analysis Can-PREDDICT	eGFR 15–45, measured hscTnT and NT-proBNP	Patients with a life expectancy <12 months, active vasculitis, or organ transplantation
Iwaski	2016	Japan	149 (CKD 1–2 definition unclear)	Japanese equation eGFR < 60 mL/min/1.73 m^2^	56 (Including CKD 1–2)	Mean 74 (Including ckd 1–2)	Single-center prospective cross-sectional study	Patients with chest symptoms	Data unavailability
Miller-Hodges & Anand	2018	Scotland	904	MDRD eGFR < 60 mL/min/1.73 m^2^	48	Mean 77	Prospective multicenter study	All patients in whom the attending physician requested cTn for suspected ACS, at least one creatinine measurement during index presentation	STEMI, not living in Scotland
Twerenbold 0/1	2018	(APACE)	487	EPI formula eGFR < 60 mL/min/1.73 m^2^	58	Median 79	Prospective multicenter study	Adult patients presenting to the ed with symptoms suggestive of AMI (e.g., acute chest discomfort and angina paectoris) with an onset or peak within the last 12 h	Dialysis patients, STEMI, patients in whom the final diagnosis remained unclear, patients with no available hs-cTnT or hs-cTNI concentrations determined on presentation to the ED and after 1 h
Chotivanawan	2012	Thailand	89	Cockcroft–Gault formula additional analysis MDRD eGFR < 60 mL/min/1.73 m^2^	58	Mean 67	Prospective single-center study	Patients with CKD without history of MI within 14 days	History of angina pectoris or heart failure that may be angina equivalences, burn, acute neurological disease such as cerebral infarction or intracranial hemorrhage, severe sepsis, acute pulmonary embolism, pulmonary hypertension, myocarditis, pericarditis, tachyarrhythmias, receiving chemotherapy, or chest trauma
Twerenbold	2015	Europe (APACE)	447	MDRD < 60 mL/min/1.73 m^2^	56	Median 77	Prospective multicenter study	Adult patients presenting to the ED with symptoms suggestive of AMI (e.g., acute chest discomfort and angina pectoris) with an onset or peak within the last 12 h	Terminal kidney failure, no creatinine or cTn measurement were taken or the final diagnose was unclear
Sitthichanbuncha	2015	Thailand	210	EPI formula eGFR < 60 mL/min/1.73 m^2^	60	Mean 71	Single-center prospective study	Admitted to ED with chest pain, hs-TnT results and coronary angiographic results after 2 h of chest pain	Hemodialysis, inappropriate time of hs-TnT level, stress-induced cardiomyopathy, and pulmonary embolism

**Table 2 diagnostics-12-00276-t002:** Troponin cut-off details for the included studies for the systematic review of cut-off values for troponin in patients with chronic kidney disease.

	Troponin Assay	Manufacture	Suggested Cut-Off	Manufacture URL	Sensitivity %	Specificity %	Statistical Analysis	Endpoints
Soeiro	cTnI	Siemens	5.1 ng/L	40 ng/L	80.60%	42%	ROC 95% confidence interval	NSTEMI (significant coronary lesion (>70%) viewed with coronary angiography or cardiac MR)
Sukonthasarn	cTnT	Roche	100 ng/L	14 ng/L	90.90%	84,50%	ROC	AMI
Chenevier-Gobeaux	hS-cTnT	Roche	35.8 ng/L for AMI 43.2 ng/L for NSTEMI	14 ng/L	AMI = 94 NSTEMI = 92	AMI = 86 NSTEMI = 88	ROC Kruskal–Wallis for multiple comparisons	AMI (STEMI and NSTEMI combined) and NSTEMI (separately)
Kraus	(1) hs-cTnI (2) hs-cTnT	(1) Abbot (2) Roche	(1) 54.0 ng/L (2) 50 ng/L	(1) 30 ng/L (2) 14 ng/L	hs-cTnI = 82 hs-cTnT = 66	hs-cTnI = 90 hs-cTnT = 80	ROC (used an algorithm that worked better)	NSTEMI
Ryu	cTnT	Roche	350 ng/L	14 ng/L	95	97	ROC	AMI
Lim	hs-cTnI	Siemens	HD = 75 ng/L PD = 144 ng/L	47.34	hd = 93.3 pd = 100	hd = 60.76 pd = 83.1	ROC	STEMI and NSTEMI
Yang	hs-cTnT	Roche	CKD3-5 = 129 ng/L CKD3 = 99.55 CKD4 = 129.45 CKD5%D = 105.5 CKD5wD = 149.35	14 ng/L	total 75.2 CKD3 = 82.8 CKD4 = 73.2 CKD5%D = 81 CKD5wD = 79.2	total 83.2 CKD3 = 82.1 CKD4 = 85.4 CKD5%D = 88.9 CKD5wD = 81.9	ROC	STEMI and NSTEMI
Flores-Solís	cTnI	(1) Beckman Coulter (2) BioMérieux	(1) 110 (2) 60	(1) 40 ng/L (2) 110 ng/L	(1) = 68% (2) = 75%	(1) = 83% (2) = 79%	ROC	ACS
Canney	hs-cTnT	Roche	30–44 mL/min = 22.7 ng/L 20–29 mL/min = 26.8 ng/L <20 mL/min = 35.5 ng/L	14 ng/L	NA	NA	Repeated log-rank test	CV risk in asymptomatic patients.
Iwasaki	cTnT	Roche	CKD 1–2 = 47 ng/L CKD 3 = 88 ng/L CKD 4–5 = 180 CKD5d = 270	14 ng/L	CKD 1–2 = 59.3 CKD 3 = 55.6 CKD 4–5 = 66.7 CKD 5d = 64.3	CKD 1–2 = 80.0 ng/mL CKD 3 = 92.2 CKD 4–5 = 89.7 CKD 5d = 78.8	ROC	ACS
Miller-Hodges & Anand	(1) cTnI (2) hs-cTnI	(1) Abbot (2) Abbot	<5 ng/L for hs-cTnI a threshold for risk stratification from earlier studies by the authors	16 ng/L in women 34 ng/L in men	98.9	22.8	Evaluation of sensitivity, specificity for the 5 ng/L threshold	NSTEMI or cardiovascular death within 30 days
Twerenbold 0/1	(1) hs-cTnT (2) hs-cTnI	(1) Roche (2) Abbot	(1) Rule-in 0 h ≥ 52 ng/L OR 1 h–change ≥5 ng/L (2) Rule-in 0 h ≥ 52 ng/L OR 1 h–change ≥ 6 ng/L	(1) 14 ng/L (2) 26.2 ng/L	hs-cTnT rule out 100% rule out hs-cTnI 98.6	hs-cTnT rule in 88.7% rule in hs-cTnI 84.4	ROC comparison of ROC	NSTEMI
Chotivanawan	hs-cTnT	Roche	CKD 3–5 0.139 ng/mL stage 3 = 0.052 stage 4 = 0.136 stage 5 = 0.297 ng/mL	14 ng/L	NA	NA	Mean	Mean of hs-cTnT in asymptomatic patients
Twerenbold	(1) hs-cTnT (2) cTnI (3) hs-cTnI (4) cTnI (5) hs-cTnI (6) c TnI (7) hs-cTnI	(1) Roche (2) Abbott (3) Abbot (4) Siemens (5) Siemens (6) Beckman-Coulter (7) Beckman-Coulter	(1) = 29.5 ng/L, (2) = 27 ng/L (3) = 29.4 ng/L, (4) = 46 ng/L, (5) = 32.0 ng/L (6) = 36 ng/L (7) = 25.9 ng/L	(1) = 14 ng/L (2) = 28 ng/L (3) = 26.2 ng/L (4) = 40 ng/L (5) = 9.0 ng/L (6) = 42 ng/L (7) = 9.2	(1) = 84 (2) = 79 (3) = 76 (4) = 77 (5) = 82 (6) = 81 (7) = 81	(1) = 79 (2) = 87 (3) = 85 (4) = 84 (5) = 83 (6) = 88 (7) = 83	ROC	AMI
Sitthichanbuncha	hs-cTnT	Roche	41 ng/L	14 ng/L	CKD3: 67 CKD4-5: 71		ROC	Coronary artery occlusion 70%

**Table 3 diagnostics-12-00276-t003:** Different troponin cut-offs established by meta-analysis for the systematic review of cut-off values for troponin in patients with chronic kidney disease.

Outcome	Cut-Off (95%CI)	Sensitivity (95%CI)	Specificity (95%CI)	AUC	(95%CI)
All	72.6 (25.75; 119.45)	0.78 (0.52; 0.93)	0.84 (0.62; 0.96)	0.89	(0.79; 0.96) * (0.73; 0.91) ^+^
TNT without dialysis	47.89 (23.95; 71.83)	0.76 (0.52; 0.92)	0.78 (0.54; 0.92)	0.85	(0.72; 0.95) * (0.65; 0.87) ^+^
TNT with dialysis	239.75 (69.27; 410.23)	0.88 (0.48; 0.99)	0.92 (0.6; 1)	0.96	(0.85; 1.00) * (0.79; 0.95) ^+^
TNI without dialysis	42.45 (33.83; 51.08)	0.77 (0.72; 0.81)	0.86 (0.84; 0.88)	0.89	(0.86; 0.92) * (0.88; 0.91) ^+^

Abbreviations: CI (confidence interval), AUC (area under the curve). *: confidence interval calculated for sensitivity given specificity, ^+^: confidence interval calculated for the specificity, given sensitivity.

## Data Availability

Data sharing not applicable. No new data were created or analyzed in this study. Data sharing is not applicable to this article.

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
