# Peer review of "Troponin Cut-Offs for Acute Myocardial Infarction in Patients with Impaired Renal Function—A Systematic Review and Meta-Analysis"

_diagnostics, 2022, doi:10.3390/diagnostics12020276_

Round 1

Reviewer 1 Report

In this manuscript the author aimed to identify specific troponin T/I values for diagnosis of AMI in patients with renal impairment via meta-analysis. After an initial screening of a total of 2,590 publications, the author extracted 15 eligible studies by inclusion and exclusion criteria. Then these studies were assessed for quality and bias of diagnostic accuracy with QUADAS-2 score system. Finally 6 studies were considered to be adequate and selected for the meta-analysis.

The methods used in the study were almost appropriate. The findings are of interest and contain robust clinical implication.

At first, the author should mainly focus on the final 6 studies and meta-analysis for discussion about determining adequate threshold of troponin once after assessments with QUADAS-2 score sytem for 15 eligible studies. These other 9 studies could not contribute to the estimation of troponin cut-offs established by meta-analysis.

The definitions of AMI are heterogeneous and the definitions of renal impairment are also heterogeneous in the included studies. Those affect quality of patient selection, indeed QUADAS-2 showed high risk score in patient selection. Especially, the definition of AMI as endpoint in included studies are crucial because some include only NSTEMI, other include NSTEMI and STEMI, also each study uses each guideline for diagnosis of AMI. In 5 studies except a study, the cardiologists are not blinded for the troponin results, which means patient selection itself were severely affected by troponin results as index examinations. This is the very serious limitation of this meta-analysis study.

Because of the number of cut-off values, study by Twerenbold, et. al. has great impacts on determining cut-off value of troponin, it looks like a dominant study in this meta-analysis. For example, SROC curve for the troponin cut-off in patients with CKD is almost identical of ROC with Twerenbold’s study alone in figure 4. This study examined 7 different assay system with each 6 (5) cut-off values. The cut-off values of troponin were determined for an each subcohort of each assay but study population are not independent for each assay. Is this fair to treat the cut-off values from the 7 assays of this study from some non-redundant patients as independent result? Also in dialysis patients, the cut-off values from SROC curve of meta-analysis almost identical with the value of study by Ryu, et. al. The cut-off value for AMI diagnosis in non-dialysis or dialysis patients determined in the meta-analysis as the main result of this study seems to come from the results of these 2 dominant studies which has unique definitions of AMI and renal dysfunction than other study included in this meta-analysis and this is a serious limitation of this study as the author mentioned in the discussion.

The author mentioned about ethnic differences in cardiac troponin value and also emphasized in the conclusion. But the studies the author included are one or two with somewhat different definition of AMI and renal impairment. So this is not scientific way of speculation and conclusion.

Author Response

Dear reviewer, thank you very much for your comments and suggestions. The point-by-point response is written in red letters.

Reviewer 1

In this manuscript the author aimed to identify specific troponin T/I values for diagnosis of AMI in patients with renal impairment via meta-analysis. After an initial screening of a total of 2,590 publications, the author extracted 15 eligible studies by inclusion and exclusion criteria. Then these studies were assessed for quality and bias of diagnostic accuracy with QUADAS-2 score system. Finally 6 studies were considered to be adequate and selected for the meta-analysis.

The methods used in the study were almost appropriate. The findings are of interest and contain robust clinical implication.

At first, the author should mainly focus on the final 6 studies and meta-analysis for discussion about determining adequate threshold of troponin once after assessments with QUADAS-2 score sytem for 15 eligible studies. These other 9 studies could not contribute to the estimation of troponin cut-offs established by meta-analysis.

Dear reviwer 1,

Thank you very much for your thorough review of our manuscript.  I can see that the differing number of papers included in the review and the meta analysis can cause some confusion.

The decision to keep the 9 studies not included in the meta-analysis in the review  was to illustrate the whole spectrum of studies on the topic. Some studies (e.g. Miller-Hodges et al. High-Sensitivity Cardiac Troponin and the Risk Stratification of Patients With Renal Impairment Presenting With Suspected Acute Coronary Syndrome and Twerenbold et al. 0/1-Hour Triage Algorithm for Myocardial Infarction in Patients With Renal Dysfunction) were well designed high quality studies, but did not fit in the meta-anlysis due to their respective study designs. Since the study is both a review and a meta-analysis, we believe that it is important also to show these data. In order to improve readability, we sub-divided results and discussion into a systematic review with a narrative synthesis of the 9 studies and a meta-analysisof the remaining 6 studies. (Under results p. 3 line 135-140, p. 22 line 310; Under discussion p. 26 line 363, p.27 line 424 highlighted in turquoise).

The definitions of AMI are heterogeneous and the definitions of renal impairment are also heterogeneous in the included studies. Those affect quality of patient selection, indeed QUADAS-2 showed high risk score in patient selection. Especially, the definition of AMI as endpoint in included studies are crucial because some include only NSTEMI, other include NSTEMI and STEMI, also each study uses each guideline for diagnosis of AMI. In 5 studies except a study, the cardiologists are not blinded for the troponin results, which means patient selection itself were severely affected by troponin results as index examinations. This is the very serious limitation of this meta-analysis study.

Indeed, we also agree that patient selection and definition of AMI and renal impairment is a limitation of this study, as mentioned in the discussion (p. 26 line 368-369 highlighted in turquoise). This is a central problem when dealing with studies on AMI and CKD. Our review calls for further studies and which we address in this study and whope that by presenting the present literature and its challenges will highligt this problem and  may lead to improved  study designs in the future. We have reiterated this point  on p. 28 line 479-480. We agree that a subdivision in patients with NSTEMI or STEMI would have been important since Troponin is most useful in the diagnosis of NSTEMI. However, this subdivision was not possible with the included publication. This again would be important for future studies.

Because of the number of cut-off values, study by Twerenbold, et. al. has great impacts on determining cut-off value of troponin, it looks like a dominant study in this meta-analysis. For example, SROC curve for the troponin cut-off in patients with CKD is almost identical of ROC with Twerenbold’s study alone in figure 4. This study examined 7 different assay system with each 6 (5) cut-off values. The cut-off values of troponin were determined for an each subcohort of each assay but study population are not independent for each assay. Is this fair to treat the cut-off values from the 7 assays of this study from some non-redundant patients as independent result? Also in dialysis patients, the cut-off values from SROC curve of meta-analysis almost identical with the value of study by Ryu, et. al. The cut-off value for AMI diagnosis in non-dialysis or dialysis patients determined in the meta-analysis as the main result of this study seems to come from the results of these 2 dominant studies which has unique definitions of AMI and renal dysfunction than other study included in this meta-analysis and this is a serious limitation of this study as the author mentioned in the discussion.

We agree that the amount of comparable high quality publications is a major limitation. Despite the small amount of papers, we were able generate cut-offs with acceptable sensitivity and specificity thanks to the amount of different cut-off values with multiple cut offs per publication. The featured tables in the manuscript leave it to the reader to compare different assays and different CKD stages according to the different studies and make a subjective personal review of the available cut-off feasible. The Twerenbold study scored very high in the QUADS2 score and we, therefore believe that the dominance of the Twerenbold study material is not necessarily a disadvantage in itself, however we agree it is a potential limitation (see p. 28 line 452-454 highlighted in turquoise). As for the 7 different TnI assays, interestingly the TnI assays had very similar cut-offs despite the URL being different in the assays. We, therefore, would argue that using 7 assays under one umbrella is acceptable (see p. 28 line 454-456 highlighted in turquoise). The overall high specificity and sensitivity makes us confident that despite a small amount of available publications our optimized cut-offs are sound. However, further research is necessary to support these findings. The dilemma regarding definitions of diagnoses has been addressed under the previous comment.  

The author mentioned about ethnic differences in cardiac troponin value and also emphasized in the conclusion. But the studies the author included are one or two with somewhat different definition of AMI and renal impairment. So this is not scientific way of speculation and conclusion.

Thank you for pointing this out. We have now removed this text from the manuscript.

Reviewer 2 Report

This study is very well written and the subject is very important. The idea behind the study is good. However, there is no clear conclusion. I understand the study question, but there are no clear clinical take home messages.

Author Response

Dear reviewer, thank you very much for your comments and suggestions. See our point-by-point response in red letters beneath.

Reviewer 2

This study is very well written and the subject is very important. The idea behind the study is good. However, there is no clear conclusion. I understand the study question, but there are no clear clinical take home messages.

Dear reviewer 2,

Thank you very much for your kind comments.

We do agree that the conclusion and take home message mightappear inconclusive. Our review was designed to gather the available information on troponin cut-offs. Furthermore, we attempted to establish an optimized cut-off value with a convincing specificity and sensitivity. These cut-offs were based on the available literature, which as you have pointed out, is sparse Therefore, many subgroup analyses were not possible and we were limited by the quality of the available publications. Our results are therefore, tentative and require further investigation, as we point out in our discussion and conclusion. We hope that the issues mentioned in the review regarding the study design of the included studies might improve further research as we address the quality and design of included studies. We added this suggestion on p. 28 line 479-480.

We are careful not to present our results as ultimate troponin cut-off, but more as a suggestion or reference for clinicians, as no other systematic reviews in this area have collated the available data prior to this review. We believe that our tables might function as a possibility for clinicians to review the available cut-off values in the literature. However, as we mention in our manuscript, an optimized cut-off will never substitute a thorough anamnesis and ECG study, but should rather be seen as a diagnostic aid. Therefore, our conclusions are to a certain point our proposed optimized cut-off, based on the current available literature. Our take home message is the call for further studies on this subject, as we have identified a potential gap in research. We have addressed this point in the conclusion and amended our conclusion accordingly p.29 line 489-491

Reviewer 3 Report

Your  researches end in february 2021: could you continue it in this time-lapse, or simply add in the discussion that a rapid review of the last scientific  contributions excluded substantial modifications?

You could report the attitude for troponin be cleared during hemodyalysis, so to indicate as a diagnostic index the persistence or increase in its value.  It would be also interesting to know  the behaviour of this enzymes in course of acute kidney failure, mainly in course of shock.  

Author Response

Dear reviewer, thank you very much for your comments and suggestion. See our answers in red letters.

Reviewer 3

Your researches end in february 2021: could you continue it in this time-lapse, or simply add in the discussion that a rapid review of the last scientific  contributions excluded substantial modifications?

Dear Reviewer,

Thank you very much for your important comments and suggestions. We have discussed the subject of a rapid review in the research group and find the suggestion of the same could have relevance to this study. However, our experience of rapid reviews is that they are frequently not so rapid and of inferior quality. Therefore, we have decided not to perform a rapid review.

You could report the attitude for troponin be cleared during hemodyalysis, so to indicate as a diagnostic index the persistence or increase in its value.  It would be also interesting to know  the behaviour of this enzymes in course of acute kidney failure, mainly in course of shock.  

We will attempt to respond to our understanding of this comment. Should we have misunderstood the point of this comment, please send your comment again.   

The influence of hemodialysis on troponin is an important factor. Studies have shown increase, decrease and no change of troponin following dialysis. No study including dialysis patients from this review described the time point of troponin measurement (p.26 line 375-377 highlighted in turquoise). The troponin fluctuation in acute kidney failure is another interesting and important point. Most studies from this review have not expanded on whether patients had a normal renal function prior to their involvement in the study or not (p. 26 line 371-374 highlighted in turquoise).

These are interesting points and require further research. However, to discuss them in depth in this review would alter the focus of the review but we do encourage further studies in this area. Therefore, we have broached the topic briefly only in the discussion section.

Round 2

Reviewer 1 Report

No additional comments for the revised version of this manuscript.